# Predictive Analyses of Prognostic-Related Immune Genes and Immune Infiltrates for Glioblastoma

**DOI:** 10.3390/diagnostics10030177

**Published:** 2020-03-24

**Authors:** Ping Liang, Yi Chai, He Zhao, Guihuai Wang

**Affiliations:** 1School of Clinical Medicine, Tsinghua University, Beijing 100084, China; liangp17@mails.tsinghua.edu.cn (P.L.); chaiy18@mails.tsinghua.edu.cn (Y.C.); 2School of Materials, Tsinghua University, Beijing 100084, China

**Keywords:** glioblastoma, prognosis, immune-related, prediction value

## Abstract

Glioblastoma (GBM), the most common and aggressive brain tumor, has a very poor outcome and high tumor recurrence rate. The immune system has positive interactions with the central nervous system. Despite many studies investigating immune prognostic factors, there is no effective model to identify predictive biomarkers for GBM. Genomic data and clinical characteristic information of patients with GBM were evaluated by Kaplan–Meier analysis and proportional hazard modeling. Deseq2 software was used for differential expression analysis. Immune-related genes from ImmPort Shared Data and the Cistrome Project were evaluated. The model performance was determined based on the area under the receiver operating characteristic (ROC) curve. CIBERSORT was used to assess the infiltration of immune cells. The results of differential expression analyses showed a significant difference in the expression levels of 2942 genes, comprising 1338 upregulated genes and 1604 downregulated genes (*p* < 0.05). A population of 24 immune-related genes that predicted GBM patient survival was identified. A risk score model established on the basis of the expressions of the 24 immune-related genes was used to evaluate a favorable outcome of GBM. Further validation using the ROC curve confirmed the model was an independent predictor of GBM (AUC = 0.869). In the GBM microenvironment, eosinophils, macrophages, activated NK cells, and follicular helper T cells were associated with prognostic risk. Our study confirmed the importance of immune-related genes and immune infiltrates in predicting GBM patient prognosis.

## 1. Introduction

Glioma is the most common cancer in the central nervous system (CNS), and about half of patients present with a highly aggressive form of glioblastoma (GBM). GBM has a poor prognosis despite maximal surgical resection and subsequent chemoradiation. To the best of our knowledge, most GBM patients will relapse after first-line treatment; the 1-year survival rate is approximately 25% and the mean 5-year survival rate is less than 3% [1,2]. Currently, pathological studies are inadequate in predicting the prognosis of patients. With the development of tumor molecular genetics, classification at the gene level can more accurately reflect disease outcome compared with histological classification [3]. For example, IDH (Isocitrate Dehydrogenase) mutation has been included in the clinical diagnostic criteria of gliomas, which contributes to the prediction of GBM prognosis. Nevertheless, there is still an urgent demand for the development of new strategies to effectively assess the GBM prognosis [4].

Intrinsic genes of tumor cells, especially master transcription factors, dominate the initiation and progression of GBM [5,6]. Additionally, the tumor microenvironment contains infiltrating immune cells that also have a significant influence on tumor gene expression, which affect clinical outcomes [7,8]. The GBM microenvironment is generally immunosuppressive and contains infiltrating immune cells, including microglia and macrophages, NK cells, and neutrophils, but a paucity of T cells and non-immune components, such as astrocytes, neurons, and tumor cells [9]. However, which immune genes and immune cells are associated with GBM prognosis remains unknown. Therefore, a better understanding of the underlying immune pathological mechanisms of glioblastoma progression is critical.

This study screened GBM prognosis-related immune genes from the TCGA (The Cancer Genome Atlas) and ImmPort databases and constructed prognostic models for clinical prediction. We also established a regulatory network of prognostic-related immune genes and transcription factors. As a result, we confirmed the relevance of several immune genes to GBM patient outcome, risk score, and the infiltration of immune cells.

## 2. Materials and Methods

### 2.1. Raw Data Preparation

The clinical data and transcriptome files of GBM HTSeq-Counts and HTSeq-FPKM were downloaded from the TCGA official website (available online: https://portal.gdc.cancer.gov/, accessed on 19 March 2020). The mRNA data from GENECODE (available online: https://www.gencodegenes.org/human/, accessed on 19 March 2020) was extracted for differential analysis through R language, while logFC (log fold change) >2 and FDR (false discover rate) <0.05 were used as screening conditions. The normalization value of fragments per kilobase million (FPKM) was presented as a transcripts per million (TPM) value.

### 2.2. Screening of Prognostic-Related Immune Genes

The immune-related gene list was downloaded from the ImmPort website (available online: https://www.immport.org/shared/home, accessed on 19 March 2020). With the integration with clinical data, prognostic immune genes were screened with single-factor Cox analysis. Here, *p* values <0.05 were considered as the prognostic-related immune gene, hazard ratio >1 was considered as the prognostic pathogenic gene, and 0< hazard ratio <1 was considered as the prognostic protection gene. The results were shown in a forest plot.

### 2.3. Construction of Prognostic Immune Genes and Immune Gene Transcription Factor Regulatory Networks

The immune gene transcription factors were downloaded from the Cistrome website (available online: http://www.cistrome.org/, accessed on 19 March 2020). The correlation test between prognostic immune genes and immune gene transcription factors was performed. Correlation coefficient >0.4 was regarded as positively correlated, while < −0.4 was regarded as negatively correlated. Cytoscape software was used to construct the regulatory network, and *p* values <0.05 were considered statistically different.

### 2.4. Establishment of Immune Gene Model

In our work, the immune gene model was built using the TCGA dataset (test set *n* = 132), while the CGGA (Chinese Glioma Genome Atlas) dataset was used for validation (*n* = 220). Then, univariate Cox analysis was used to filtrate genes affecting overall survival of patients (*p* < 0.05), followed by multivariate Cox analysis to identify genes as independent prognostic indicators. Subsequently, based on the expression level of each gene and the coefficient obtained from multivariate Cox analysis were used to conduct a risk score. The algorithm is shown below.

Risk score = ExpmRNA1×coefmRNA1 + ExprmRNA2×coefmRNA2+⋯+ExpmRNAn×coef mRNA, where Exp represents the expression level of each gene and coef represents the coefficient of each gene).

### 2.5. Survival Analysis

Survival analysis was performed on patients in the high-risk score and low-risk score groups, and *p* values < 0.05 were considered statistically different.

### 2.6. Receiver Operating Characteristic (ROC) Curve

To evaluate the sensitivity and specificity of the model, the ROC curve was drawn, while the area under curve (AUC) value was calculated to assess the model, with values of 0.5–0.7 indicating moderate, 0.7–0.9 indicating better, and >0.9 indicating superior values.

### 2.7. Risk Curve

To exhibit the results of the model construction, patients were ranked according to the risk score and survival time, and the heat map was drawn to show the trend of gene expression required for the model.

### 2.8. Independent Prognostic Analysis

To assess whether the model can be used as an independent factor to predict the prognosis of patients, univariate and multivariate independent prognostic analyses were performed. Age and gender were comparative factors, and age comparison was separated by 50 years.

### 2.9. Clinical Correlation Analysis

To evaluate differences of prognostic immune genes and risk scores in clinical characteristics, *p* values >0.05 were considered to be related to corresponding clinical features.

### 2.10. Correlation Analysis of Immune Cell Infiltration and Risk Score

The limma package was used to revise the original expression matrix, then CIBERSORT was used to predict the immune cell composition. Here, *p* values <0.05 were considered to be highly accurate and were used for subsequent analysis, including the correlation between the content of immune cells and the risk score, percentage of immune cells, and correlation between immune cells.

## 3. Results

### 3.1. Identification of Prognostic-Related Immune Genes

We collected a total of 161 samples from the TCGA official website, including 5 normal samples and 156 GBM samples. The vst function was used to normalize the read counts and a difference analysis was performed, then logFC was corrected by IfcShrink (Figure 1a,b). Next, we performed principal component analysis (PCA) on the samples (Figure 1c). Consequently, we obtained a total of 2942 differential expressed genes, of which 1338 genes were up-regulated and 1604 genes were down-regulated (Figure 1d,g).

Then, we downloaded 2499 immune-related genes from the ImmPort database. After the intersection with 2942 differential expressed genes, a total of 291 genes were obtained, of which 183 genes were up-regulated and 108 genes were down-regulated (Figure 1e,h). After screening these 291 immune genes, 24 immune genes related to prognosis were collected (Figure 2a).

### 3.2. Analysis of Regulatory Network of Immune Transcription Factors and Prognostic-Related Immune Genes

We also downloaded 318 immune gene transcription factors from Cistrome. After intersection with differentially expressed genes, a total of 41 immune-related transcription factor genes were obtained, of which 32 were up-regulated genes and 9 were down-regulated genes (Figure 1f,i). Then, by calculating its correlation with 24 prognostic immune genes, we constructed a regulatory network, with BATF (Basic Leucine Zipper ATF-Like Transcription Factor), SNAI2 (Snail Family Transcriptional Repressor 2), GATA4 (GATA Binding Protein 4), HOXB13 (Homeobox B13), RUNX1 (Family Transcription Factor 1), RUNX1T1 (RUNX1 Partner Transcriptional Co-Repressor 1), and WWTR1 (WW Domain Containing Transcription Regulator 1)found to have certain regulatory effects on related genes (Figure 2b, Table 1).

### 3.3. Validation of Prognosis Prediction Model and Survival Analysis

We modeled 24 prognostic-related immune genes, from which 9 genes were chosen for modeling (Table 2). To evaluate the sensitivity and specificity of the model, we drew an ROC curve, for which the area under the curve (AUC) was 0.869, which showed the relative authenticity of the model (Figure 3a). The Kaplan–Meier survival curve showed statistically significant differences between high- and low-risk patients (*p* < 0.001). Notably, patients in the low-risk group had significantly longer survival time (Figure 3b). The survival rate corresponding to each time point is shown in Table 3. The validation results were similar in survival analysis (*p* < 0.05, more details in Appendix A).

### 3.4. Risk Curve Analysis

According to the risk curve, it is obvious that the survival of low-risk patients is relatively longer than that of high-risk group (Figure 4a,b). The amount of each gene expression needed to construct the model is shown in the heat map. With the rise of the risk score, the expression of FABP5 (Fatty Acid Binding Protein 5), BMP1 (Bone Morphogenetic Protein 1), and OSMR (Oncostatin M Receptor) gradually increased, while the expression of CCL1 (C-C Motif Chemokine Ligand 1) and LPA (Lipoprotein(A)) decreased (Figure 4c).

### 3.5. Univariate and Multivariate Independent Prognostic Analysis

To study the prognosis-related factors, we performed univariate and multivariate independent prognostic analyses on the model. It can be seen that the univariate (Figure 5a) and multivariate (Figure 5b) analysis results are consistent. Both age and risk score can be taken as independent predictors, while gender cannot.

### 3.6. Clinical Relevance Verification

We also studied the correlation of immune genes and risk scores with clinical characteristics. We found that BMP1 and OSMR scores for all GBM patients were closely related to age, and that the expression of these two genes was significantly higher at >50 years old (Figure 6a,b). Meanwhile, CCL1 score was related to gender, the expression of which was higher in male patients than females (Figure 6c). Patients over 50 had higher risk scores, while patients under 50 had lower risk scores (Figure 6d). The remaining genes were not found to be related to clinical characteristics.

### 3.7. Correlation between Immune Cell Infiltration and Risk Score

We evaluated the impacts of different immune infiltrates on GBM clinical outcomes, and the infiltration of eosinophils, macrophages, activated NK cells, and T cell follicular helper was found to be associated with prognostic risk. Among them, M0 macrophages and activated NK cells were positively correlated with risk (Figure 7b,c), while eosinophils and T cell follicular helpers were negatively correlated (Figure 7a,d). No statistical significance was found between other infiltrates and the risk score.

We also screened the samples and obtained a total of 20 samples that could be well analyzed. Compared to other cells, the proportions of T cell follicular helper cells, monocytes, and M0 macrophages were relative higher (Figure 8a,b).

Meanwhile, we assessed the relevance between types of immune cells. Significantly, the relationships between CD4 naïve T cells and regulatory T cells, activated regulatory T cells and NK cells, and activated CD4 naïve T cells and NK cells were highly relevant (correlation coefficient > 0.9). Regarding the relationships between T cells and gamma delta monocytes, CD4 naïve T cells and M0 macrophages, and activated dendritic cells and resting mast cells, these three groups of cells were secondarily related (0.7< correlation coefficient <0.8) (Figure 9).

## 4. Discussion

Currently, the prognosis of GBM is very poor because of numerous complicating factors, such as degree of surgical resection, drug resistance, blood–brain barrier permeability, and radiotherapy dose selection. Even with the recent advent of tumor immunotherapy, the anti-tumor immune response is quite limited because of the difficulty of lymphocyte infiltration into the GBM microenvironment. It was reported that tumor intrinsic factors were associated with GBM microenvironment immunosuppression by inducing immune suppressive signaling pathways [10,11]. Recent molecular research has demonstrated that immune infiltration can promote and regulate tumor progression via direct interactions, although their roles in tumor origination and patient prognosis are still poorly understood. Therefore, we focused on the gene expressions of immune infiltrates and their relevance to clinical prognosis.

During the progression of GBM, some abnormally expressed genes, including immune genes, are closely related to the clinical prognosis. Therefore, it is important to determine how to identify these genes to predict the prognosis of GBM. For this, we screened prognostic immune genes and transcription factors from relevant databases to construct a regulatory network (Figure 2b). It was reported that the BATF family, which belongs to a class of transcription factors containing a basic leucine zipper domain, regulates a variety of immune functions and controls the development and differentiation of immune cells [12]. Compared with other transcription factors, we found that BATF had the strongest positive regulation on related immune genes.

After the optimization of 24 immune genes, we found that nine genes were suitable for establishing a prognostic prediction model. The AUC value of the ROC curve was 0.869, which indicated the reliability of the model. When the risk score was increased, FABP5, BMP1, OSMR2, CCL1, and LPA gene expressions also changed. In short-term survivors of GBM (≤6 months), high levels of FABP5 protein were expressed, which was associated with highly proliferating tumor cells and the activation of the v-akt murine thymoma viral oncogene homolog and 3-phosphoinositide-dependent protein kinase-1 [13]. It was previously reported that BMP1 was involved in multiple signaling pathways of GBM and was associated with a poor prognosis for glioma patients [14]. OSMR is a member of the interleukin-6 receptor family and is a key regulator of GBM growth. It regulates the feed-forward signaling pathway to promote tumorigenesis with EGFRvIII and STAT3. A correlation between the upregulation of OSMR and poor survival was previously confirmed [15]. CCL1 is an inflammatory mediator that stimulates human monocyte migration. It was reported that the introduction of CCL1 induced tumor regression and resistance against tumors [16].

Further research has indicated a correlation between four immune cell types in the tumor microenvironment and risk score. Tepper found that malignant cell lines induced the infiltration of eosinophils in a nude mouse model, leading to the attenuation of malignant tumors [17]. A clinical trial reported that the improved survival of GBM patients was linked to increased numbers of eosinophils detected post-surgery [18]. Our study revealed that an increase in infiltrating eosinophils and follicular helper T cells was associated with a decreased patient risk score, indicating an improved prognosis. However, there is little evidence that follicular helper T cells are negatively correlated with risk score. With the progress of GBM, monocytes enter the central nervous system from the periphery through the damaged blood–brain barrier. A previous study reported that the proportion of tumor-associated macrophages (TAMs) in glioma tissues reached 30% [19], and that TAMs were the core components that promoted tumor progression and immunotherapy resistance in a GBM immunosuppressive microenvironment [20]. Furthermore, a high level of macrophage infiltration in GBM resulted in a worse prognosis [21]. Our results also verified that the infiltration of M0 macrophages was positively correlated with the risk score. However, the infiltration of pro-inflammatory M1 and anti-inflammatory M2 macrophages did have a direct link to prognosis. NK cells secrete tumor necrosis factor (TNF) and interferon (IFN), which exert a killing effect on cells, as well as having an important role in anti-tumor innate immunity. Our statistical analyses also showed that activated NK cell infiltration correlated with poor prognosis. Indeed, NK activity in glioma tissues was inhibited and the number of activated NK cells was very low, which reduced the killing effect of NK cells on GBM in the GBM microenvironment [9,22,23,24].

In summary, we analyzed the prognosis of GBM patients using prognostic-related immune genes and immune infiltration, which identified predictive biomarkers and valuable clues for new immunotherapy strategies. However, there were potential limitations in our analysis. The related predictions were based on the sequencing data analysis of a public database. Furthermore, the type of immune cells was not verified by immunohistochemistry, and the results have not been verified by biological experiments. Subsequent in vivo or in vitro and clinical verification is needed to consolidate our research results.

## 5. Conclusions

In our study, relevant immune genes were identified to establish a prediction model for the prognosis of GBM patients. We also analyzed the impact of immune cell infiltration in the microenvironment on the risk score. Further research of these immune genes or immune infiltrates may provide potential therapeutic targets for GBM clinical treatment.

## Figures and Tables

**Figure 1 diagnostics-10-00177-f001:**
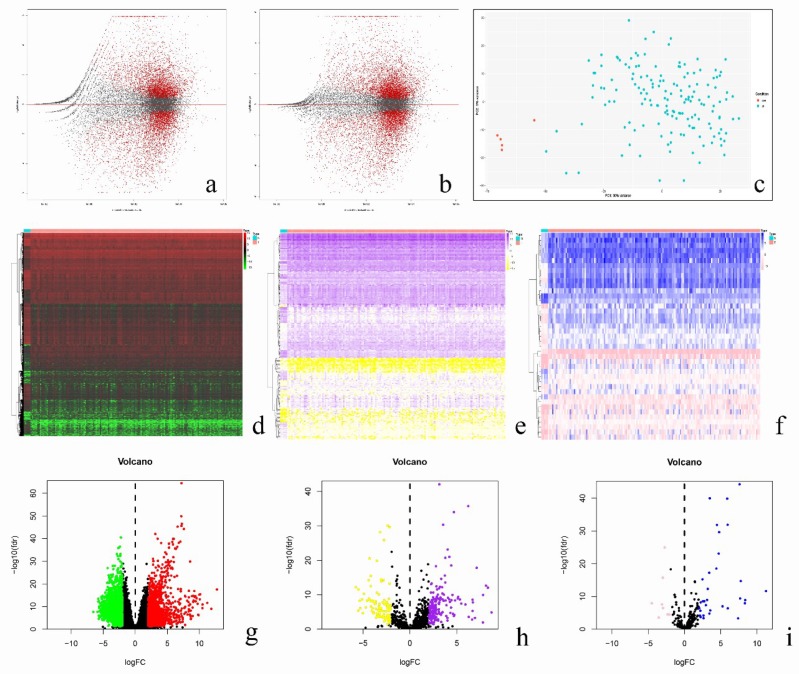
Processing and analysis of 161 glioblastoma (GBM) samples from the TCGA (The Cancer Genome Atlas) database (**a**–**c**). The heatmap and the volcano plot of 2942 differentially expressed genes among 161 GBM samples (**d**,**g**). The heatmap and the volcano plot of 291 immune-related genes obtained from ImmPort database after the intersection with differentially expressed genes (**e**,**h**). The heatmap and the volcano plot of 41 immune-related transcription factors obtained from Cistrome after the intersection with differentially expressed genes (**f**,**i**). For high-resolution images, see Appendix A.

**Figure 2 diagnostics-10-00177-f002:**
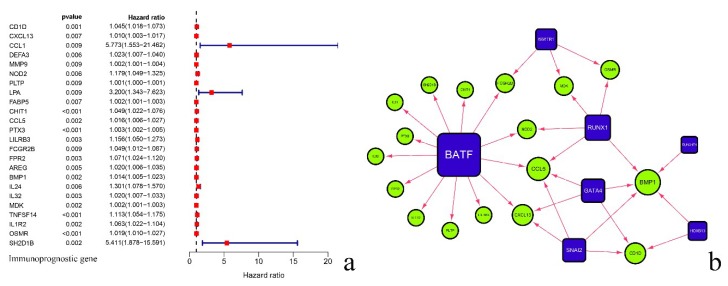
The 24 prognostic immune genes obtained after further screening (**a**), the construction of regulatory networks between immune transcription factors, and the prognostic immune gene (**b**).

**Figure 3 diagnostics-10-00177-f003:**
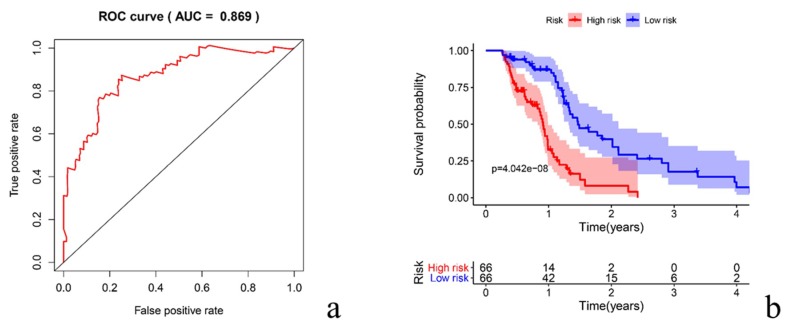
The receiver operating characteristic (ROC) curve indicates that the area under the curve (AUC) for the model is 0.869 (**a**). The K-M (Kaplan-Meier) survival curve shows the difference between the low-risk group and high-risk group patients (**b**).

**Figure 4 diagnostics-10-00177-f004:**
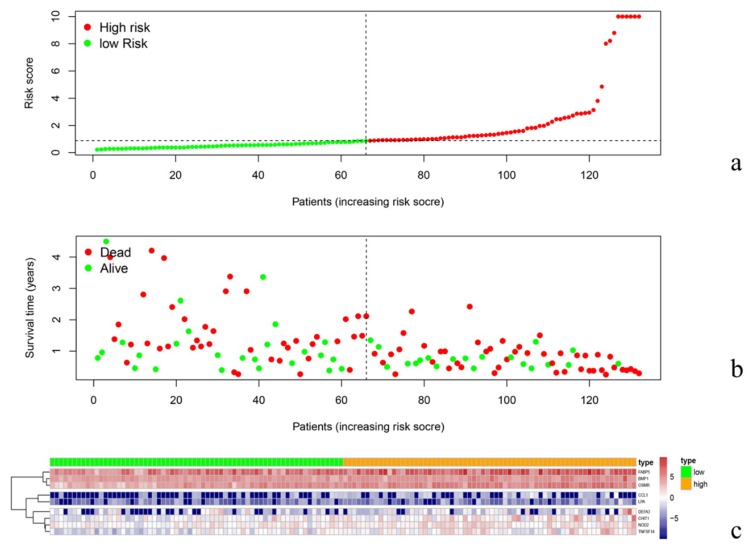
The survival was prolonged in low-risk patients compared to the high-risk group (**a**,**b**), and the changes in gene expression are shown by increased the risk score (**c**).

**Figure 5 diagnostics-10-00177-f005:**
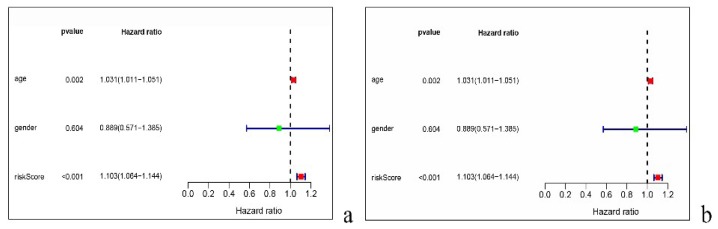
Both the age and risk score can be taken as independent predictors according to the univariate (**a**) and multivariate analyses (**b**).

**Figure 6 diagnostics-10-00177-f006:**
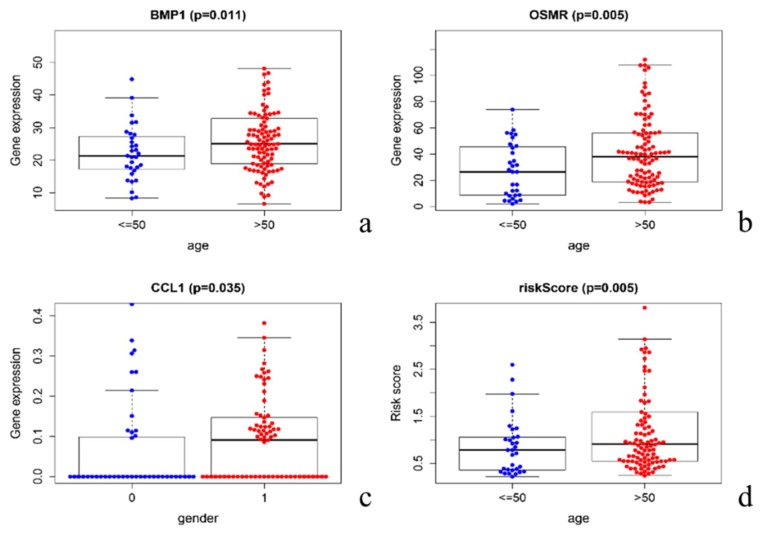
The expression of BMP1 (Bone Morphogenetic Protein 1) and OSMR (Oncostatin M Receptor) was related to age (**a**,**b**), the expression of CCL1 (C-C Motif Chemokine Ligand 1) was related to the gender (**c**), and the risk score was obviously related to age (50 years old as the age cut-off) (*p* < 0.05) (**d**).

**Figure 7 diagnostics-10-00177-f007:**
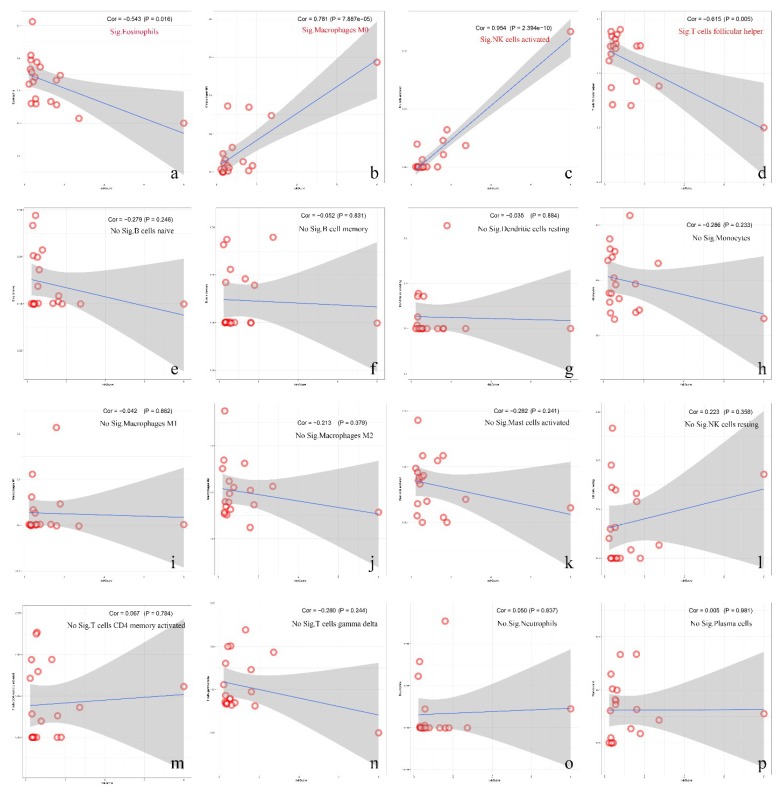
The correlations between immune infiltration cells and risk scores. Significant differences are shown in graph (**a**–**d**) (*p* < 0.05). However, in (**e**–**p**), there was no statistical significance

**Figure 8 diagnostics-10-00177-f008:**
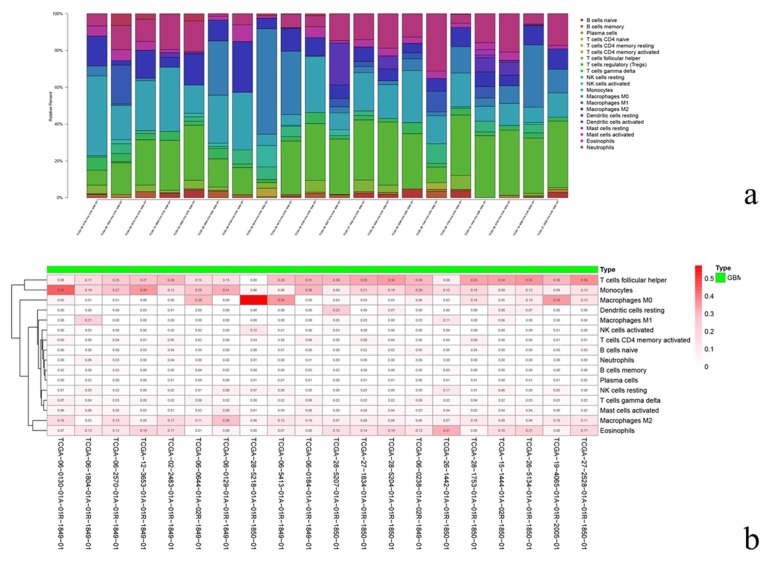
The percentages of different immune cells in the GBM microenvironment. In the 20 samples, each column represents a sample, and each column with a different color and height indicates the abundance ratios of immune cells in this sample (**a**). Each column represents a sample and each column with a different color indicates the percentage of immune cells in this sample (**b**).

**Figure 9 diagnostics-10-00177-f009:**
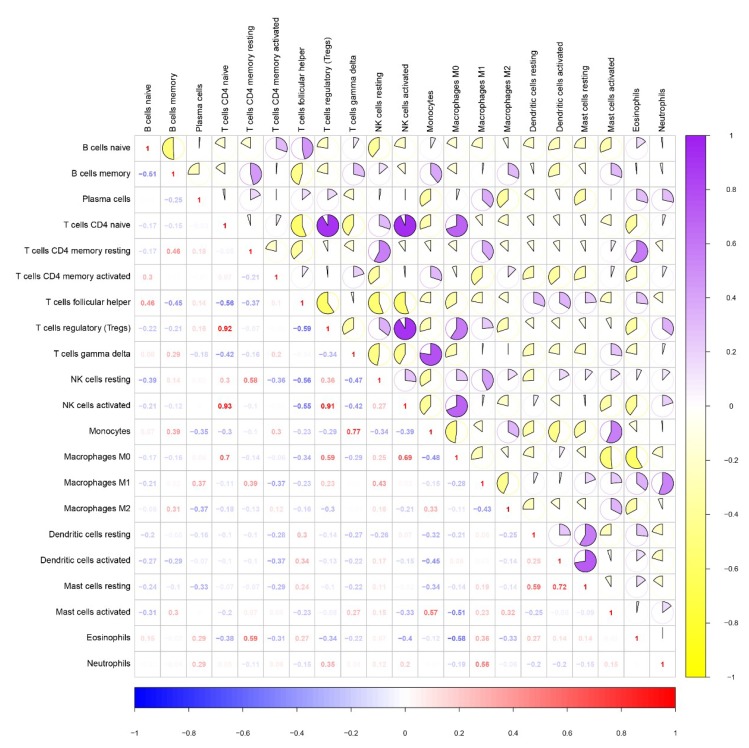
Correlation matrix of proportions of all immune cells. Some immune cells are negatively correlated (shown in yellow), while others are positively correlated (shown in purple). The darker the color, the higher the correlation (*p* < 0.05).

**Table 1 diagnostics-10-00177-t001:** Seven immune transcription factors screened for gene regulation. Detailed information of each transcriptome and its regulatory genes.

Transcription Factor	ImmuneGene	Correlation	*p* Value	Regulation	Transcription Factor	Immunegene	Correlation	*p* Value	Regulation
SNAI2	CD1D	0.70	3.15 × 10^−21^	positive	BATF	CXCL13	0.43	2.09 × 10^−7^	positive
	CXCL13	0.54	1.72 × 10^−11^	positive		NOD2	0.77	1.04 × 10^−27^	positive
	CCL5	0.63	3.57 × 10^−16^	positive		PLTP	0.73	1.20 × 10^−23^	positive
	BMP1	0.80	9.53 × 10^−31^	positive		CHIT1	0.53	5.82 × 10^−11^	positive
GATA4	CD1D	0.64	6.70 × 10^−17^	positive		CCL5	0.54	9.59 × 10^−12^	positive
	CXCL13	0.45	3.99 × 10^−8^	positive		PTX3	0.62	1.67 × 10^−15^	positive
	CCL5	0.46	1.46 × 10^−8^	positive		LILRB3	0.72	1.43 × 10^−22^	positive
	BMP1	0.64	6.94 × 10^−17^	positive		FCGR2B	0.60	1.56 × 10^−14^	positive
HOXB13	CD1D	0.55	3.04 × 10^−12^	positive		FPR2	0.71	2.94 × 10^−22^	positive
	BMP1	0.53	2.37 × 10^−11^	positive		IL24	0.48	4.35 × 10^−9^	positive
RUNX1	NOD2	0.48	3.11 × 10^−9^	positive		IL32	0.40	1.73 × 10^−6^	positive
	CCL5	0.49	1.33 × 10^−9^	positive		IL1R2	0.45	3.88 × 10^−8^	positive
	BMP1	0.55	6.08 × 10^−12^	positive		SH2D1B	0.45	2.91 × 10^−8^	positive
	MDK	0.49	1.74 × 10^−9^	positive	WWTR1	FCGR2B	0.40	1.20 × 10^−6^	positive
	OSMR	0.48	5.14 × 10^−9^	positive		MDK	0.40	1.40 × 10^−6^	positive
RUNX1T1	BMP1	0.42	4.02 × 10^−7^	positive		OSMR	0.44	1.13 × 10^−7^	positive

**Table 2 diagnostics-10-00177-t002:** Nine immune genes suitable for establishing a prognosis prediction model. Detailed information of each immune gene suitable for modeling.

ImmuneGene	coef	HR	HR.95L	HR.95H	*p* Value
CCL1	2.40	11.09	2.93	41.90	0.00
DEFA3	0.03	1.03	1.01	1.05	0.00
NOD2	−0.15	0.86	0.72	1.02	0.09
LPA	1.50	4.52	1.80	11.35	0.00
FABP5	0.00	1.00	0.99	1.00	0.05
CHIT1	0.06	1.07	1.03	1.10	0.00
BMP1	0.01	1.01	1.00	1.02	0.00
TNFSF14	0.08	1.08	1.00	1.17	0.03
OSMR	0.02	1.01	1.00	1.03	0.00

**Table 3 diagnostics-10-00177-t003:** The survival rates of high-risk and low-risk patients. Detailed survival rate corresponding to each time point.

Risk = High	Risk = Low
Time (year)	n.Risk	n.Event	Survival (%)	Std.err	Lower 95% CI	Upper 95% CI	Time (year)	n.risk	n.event	Survival (%)	Std.err	Lower 95% CI	Upper 95% CI
0.26	66.00	1.00	0.98	0.02	0.96	1.00	0.27	66.00	1.00	0.98	0.02	0.96	1.00
0.27	65.00	1.00	0.97	0.02	0.93	1.00	0.27	65.00	1.00	0.97	0.02	0.93	1.00
0.30	64.00	1.00	0.95	0.03	0.91	1.00	0.33	64.00	1.00	0.95	0.03	0.91	1.00
0.30	63.00	1.00	0.94	0.03	0.88	1.00	0.41	61.00	1.00	0.94	0.03	0.88	1.00
0.31	62.00	1.00	0.92	0.03	0.86	0.99	0.64	55.00	1.00	0.92	0.03	0.86	0.99
0.34	61.00	1.00	0.91	0.04	0.84	0.98	0.70	54.00	1.00	0.90	0.04	0.83	0.98
0.36	60.00	1.00	0.89	0.04	0.82	0.97	0.74	52.00	1.00	0.89	0.04	0.81	0.97
0.38	59.00	2.00	0.86	0.04	0.78	0.95	0.76	50.00	1.00	0.87	0.04	0.79	0.96
0.39	57.00	1.00	0.85	0.04	0.77	0.94	1.04	42.00	1.00	0.85	0.05	0.76	0.95
0.40	56.00	1.00	0.83	0.05	0.75	0.93	1.08	41.00	1.00	0.83	0.05	0.74	0.93
0.41	55.00	1.00	0.82	0.05	0.73	0.92	1.11	40.00	1.00	0.81	0.05	0.71	0.92
0.42	54.00	1.00	0.80	0.05	0.71	0.91	1.11	39.00	1.00	0.79	0.06	0.69	0.90
0.43	53.00	1.00	0.79	0.05	0.70	0.89	1.15	38.00	2.00	0.75	0.06	0.64	0.87
0.45	52.00	1.00	0.77	0.05	0.68	0.88	1.21	36.00	1.00	0.72	0.06	0.61	0.86
0.48	49.00	2.00	0.74	0.05	0.64	0.86	1.23	34.00	2.00	0.68	0.06	0.57	0.82
0.49	47.00	1.00	0.73	0.06	0.62	0.84	1.24	31.00	2.00	0.64	0.07	0.52	0.79
0.61	38.00	1.00	0.71	0.06	0.60	0.83	1.32	27.00	1.00	0.61	0.07	0.49	0.77
0.62	37.00	1.00	0.69	0.06	0.58	0.81	1.33	26.00	1.00	0.59	0.07	0.47	0.75
0.63	36.00	1.00	0.67	0.06	0.56	0.80	1.34	25.00	1.00	0.57	0.07	0.44	0.73
0.66	35.00	1.00	0.65	0.06	0.54	0.78	1.38	24.00	1.00	0.54	0.07	0.42	0.71
0.74	33.00	1.00	0.63	0.06	0.52	0.76	1.46	23.00	1.00	0.52	0.07	0.39	0.69
0.82	27.00	1.00	0.61	0.06	0.49	0.75	1.47	22.00	1.00	0.50	0.07	0.37	0.66
0.86	26.00	1.00	0.58	0.07	0.47	0.73	1.49	21.00	1.00	0.47	0.07	0.35	0.64
0.87	25.00	1.00	0.56	0.07	0.44	0.71	1.64	19.00	1.00	0.45	0.07	0.32	0.62
0.89	24.00	1.00	0.54	0.07	0.42	0.69	1.78	18.00	1.00	0.42	0.07	0.30	0.60
0.90	23.00	1.00	0.51	0.07	0.39	0.67	1.85	17.00	1.00	0.40	0.07	0.28	0.57
0.91	22.00	1.00	0.49	0.07	0.37	0.65	2.02	15.00	2.00	0.34	0.07	0.23	0.52
0.92	21.00	1.00	0.47	0.07	0.35	0.63	2.11	13.00	1.00	0.32	0.07	0.20	0.50
0.94	20.00	1.00	0.44	0.07	0.32	0.61	2.12	12.00	1.00	0.29	0.07	0.18	0.47
0.94	19.00	1.00	0.42	0.07	0.30	0.58	2.41	11.00	1.00	0.27	0.07	0.16	0.44
0.98	18.00	1.00	0.40	0.07	0.28	0.56	2.81	9.00	1.00	0.24	0.07	0.13	0.41
0.99	17.00	2.00	0.35	0.07	0.24	0.52	2.91	8.00	2.00	0.18	0.06	0.09	0.35
0.99	15.00	1.00	0.33	0.07	0.22	0.49	3.38	5.00	1.00	0.14	0.06	0.06	0.32
1.06	13.00	1.00	0.30	0.07	0.19	0.47	3.97	4.00	1.00	0.11	0.05	0.04	0.29
1.08	12.00	1.00	0.28	0.07	0.17	0.44	4.00	3.00	1.00	0.07	0.05	0.02	0.25
1.13	11.00	1.00	0.25	0.07	0.15	0.42	4.21	2.00	1.00	0.04	0.03	0.01	0.25
1.17	9.00	1.00	0.22	0.06	0.13	0.39							
1.28	8.00	1.00	0.20	0.06	0.11	0.36							
1.33	6.00	1.00	0.16	0.06	0.08	0.33							
1.50	4.00	1.00	0.12	0.06	0.05	0.30							
1.58	3.00	1.00	0.08	0.05	0.02	0.27							
2.27	2.00	1.00	0.04	0.04	0.01								
2.42	1.00	1.00	0.00

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
