# Peer review of "Predictive Analyses of Prognostic-Related Immune Genes and Immune Infiltrates for Glioblastoma"

_diagnostics, 2020, doi:10.3390/diagnostics10030177_

Round 1

Reviewer 1 Report

The manuscript is entirely based on computer simulation methods. Nothing has been proved by immunohistochemistry and/or molecular techniques which are, indeed, the valid diagnostic techniques in Pathology. Experiments with transcription factors and immune genes elecidated here, at least some, should have been done in vitro, in clell lines. Also a set of clinical samples and cell lines should have been studied in paralel with the computer based samples. Even for the prognostic conclusions. So, a big number of experiments would be required to complete the manuscript as an evaluable set.

Author Response

Responds to Reviewers
Reviewer #1:
1.
The manuscript is entirely based on computer simulation methods. Nothing has been proved by immunohistochemistry and/or molecular techniques which are, indeed, the valid diagnostictechniques in Pathology. Experiments with transcription factors and immune genes elecidated here, at least some, should have been done in vitro, in c ell lines. Also a set of clinical samples and cell lines should have been studied in paralel with the computer based samples. Even for the prognostic conclusions. So, a big number of exper iments would be required to complete the manuscript as an evaluable set.

Respond:
Dear reviewer,
Thank you for your reviewer`s professional questions. Thanks again, This manuscript does have defects and shortcomings in terms of in vitro or in vivo , and this drawback was point out in the discussion section of this manuscript . Our team have decided to focus on validation in the next research plan. In addition, h ope get your understanding and support based on the current situation in China, we could not provide in vitro or in vivo data in short term to validation . But we will attach more importance to this problem you mentioned in the future studies and will hope obtain more clinical and in vitro or in vivo data to validation for consolidating our research results
Line 255 261 page 13 14

Reviewer 2 Report

Liang P. et al developed an predictive analyses of prognostic-related immune
genes and immune infiltrates for glioblastoma. The manuscript is well-written and the methods used and is quite interesting in the context of Glioblastoma (GBM), because not much works have been done in this space.The presented results are all logical and interesting and clinically relevant. Clinical evaluation of these immune genes or immune infiltrates would be interesting to prove their predictive value.

Reviewer 3 Report

Ping Liang et al analyzed the immune related gene expression data from GBM patients. 24 genes were found to be prognostic predictor. Based on the 24-gene expression, the authors developed a rick score that accurately reflect the prognostic outcome of GBM with an AUC equals to 0.87. Furthermore, the associate of various immune components and prognostic risk were analyzed to demonstrate the vital impact of immune infiltration in GBM development. Overall, the logic and scientific rationale are clear and easy to follow. However, several key information about gene feature selection and model building is missing in the main text, which diminish the convincing level of the work. In addition, extensive typos and grammar errors were found which need detailed check to meet the publication standards.

Figure 1. The figure legend of each figure legend is too small to read. Figure 1 A and B. What is the meaning of these two figures? What is the advantage of using lfcshink? Figure 1 G, H, I. Change “volcano” to other more meaningful subtitle. Figure 2B. If all regulation is positive, then there is no need to show it on the plot. Table 1. We don’t need so many decimal point for correlation. Round it to 2 would be enough. Table 3 can be a supplement table. It is not very helpful to put it in the main text. Line 132. What is the selection criteria of the 9 genes? Why select 9 out of 24 rather than all 24 genes? What is the formulation of the risk score? What algorithm used to build the model? How did you perform the validation? Is it based on the same dataset or different dataset? Line 132. Change “finally” to “and”. “authenticity” is not an appropriate word to describe the accuracy of the model. Where did you use the GENCODE data? Did you include the master transcription factor genes in your model? If not, what is the reason? Did you observe the same prognostic effect of these master transcription factor genes with their target genes? Line 107. 1878 and 1604 does not sum up to be 2942 genes. Line 107. 110. Change “differential gene” to “differential expressed genes”. Line 14. Change “hazard modeling” to “proportional hazard modeling”. Line 109. Duplicated “Downloaded”. Line 112. remove “about”. 24 is an accurate number. Line 121. Change “transcription gene” to “transcription factor gene”. Line 145. Change “analysi” to “analysis”. Line 55. Data, not “datas”. Font size on page 8 is larger. Line 157. Change “while gender not” to “while gender can not”. Line 166. “All patients had a lower risk score at 167 age < 50 years, while a higher risk score at > 50 years (Fig 6d).” This is not an accurate description of figure 6d. Boxplot only show the difference in population median level. Line 246. “Correlated with prognosis”. Is activated NK cell infiltration correlated with better or worse prognosis?

Round 2

Reviewer 1 Report

The  manuscript has been changed according to the reviewers' comments.

Author Response

Thankyou very much~

Reviewer 3 Report

  1. I am confused about Figure 3a, and ROC plot in question 10. What are you predicting? What is the input and output? Are you using the 9 gene expression to predict patient risk level? Since the risk level is defined by these 9 genes, it make no sense to do such prediction or validation.
  2. To clarify question 9, I am asking the predictive model algorithm that you used to predict risk level. I believe it should be a classification model rather than regression because you use ROC curve to show the results.
  3. Method section should include more details about feature selection, model building, and model validation. E.g. risk score formula, algorithm used in predictive model building.

Round 3

Reviewer 3 Report

All my concerns have been addressed.